# Analysis of the Impact of Rubber Recyclate Addition to the Matrix on the Strength Properties of Epoxy–Glass Composites

**DOI:** 10.3390/polym15163374

**Published:** 2023-08-11

**Authors:** Daria Żuk, Norbert Abramczyk, Adam Charchalis

**Affiliations:** Faculty of Marine Engineering, Gdynia Maritime University, 81-225 Gdynia, Poland; n.abramczyk@wm.umg.edu.pl (N.A.); a.charchalis@wm.umg.edu.pl (A.C.)

**Keywords:** composites, static tensile test, impact strength, damage kinetics, rubber recyclate, statistical analysis

## Abstract

Currently, there is a noticeable trend of modifying new materials by using additives from the recycling of harmful waste. This is to protect the environment by using waste to produce composites and at the same time to reduce the cost of their production. The article presents an analysis of the impact of the use of rubber recyclate obtained from the utilization of car tires as a sandwich layer of epoxy–glass composites and its impact on the strength parameters of the composite. The presented research is an extension of the previously conducted analyses on composite materials modified with the addition of rubber recyclate. The four variants of the materials produced contained the same percentage amount of rubber recyclate, but differed in the way it was distributed and the number of layers. Static tensile tests as well as impact strength and kinetics of damage to samples made with and without the addition of recyclate were carried out. Observation of the structures of the materials with the use of SEM was also performed. A significant influence of the method of distributing the recyclate in layers on the strength parameters of the materials was found. In the case of composites with three and two sandwich layers of recyclate, more favorable results were obtained compared to the blank sample. In addition, the values of the impact strength measurements were subjected to statistical analysis at the significance level of α = 95%. The distributions were tested for normality with the Shapiro–Wilk test, differences between pairs were tested with the Student’s *t*-test for dependent groups, and ANOVA differences were tested for independent groups. Using the Student’s *t*-test, it was confirmed that between the pairs of variables in the configurations reference sample and modified sample, there were significant statistical differences in the distribution of impact strength measurement results for all the analyzed materials. Statistical analysis showed a significant usefulness in the selection of the material with the best strength parameters and a significant role of statistical methods in the study of anisotropic materials.

## 1. Introduction

In Poland, approximately 800,000 tons of polymer waste are generated annually. A special case of such waste is rubber waste, including tires, which accounts for about 150,000 tons per year [1]. The technology of their production is based on the vulcanization process, which is an irreversible process. A return to the initial state to enable reprocessing with the current technology and available recycling methods is not entirely achievable. In the European Union, about 80% of post-service rubber products are used tires, the composition and construction of which make recycling much more difficult than in the case of other materials such as metals or glass [2]. Legal regulations impose a constant increase in the use of these raw materials, but it is incomparably lower than metals, glass, and paper. An easily available material is ground rubber waste in the form of rubber recyclate, which can be used as a filler in new composite materials [3,4,5,6,7,8].

Due to problems in the processing of tires, most often their development ends with storage, which is an uneconomical and unecological solution. Recycling is also influenced by fillers added to polymeric materials to improve their processing parameters and the properties of the obtained products [9]. Table 1 presents the supply of used tires in the EU countries and Poland, taking into account the development of the automotive industry. Table 2 presents the quantities of tires produced and introduced in the Polish market.

Rubber recyclate is a generally available material, used in the production of rubber materials such as mats, floor tiles, carpet underlays, etc., where this material is joined using polyurethane or latex adhesives [11,12,13]. In the tire industry, this waste is also used in compounds; however, this amount will not increase due to concerns about tire performance and safety [14].

Recovered products from rubber recycling can be used in the production of new materials (e.g., production of a new product as a result of simple thermo-mechanical treatment by reducing the size and melting the resulting powders with thermoplastic resins to produce TPE thermoplastic elastomer compounds). TPEs are multifunctional polymeric materials that combine the processability of thermoplastics and the flexibility of rubbers. However, these materials are characterized by poor mechanical strength due to the incompatibility and immiscibility of most polymer blends [15]. Hence, the main problem associated with the production of TPE from recycled materials by melt blending is the low affinity and interaction between the thermoplastic matrix and the cross-linked rubber [16]. This leads to phase separation and poor adhesion between the two phases.

As part of extensive research, the use of used rubber as a binder (e.g., elastomers, bitumens) or as conglomerates (cement, gypsum) for the production of innovative composites used in construction was investigated [12,17,18]. To improve the properties of composites made of recycled rubber, the rubber surface has been subjected to various costly processes to improve the interfacial transition zone [19,20]. It was found that knowledge of the chemical, physical, and mechanical properties of rubber as well as proper characterization is essential for the full use of this material.

The research in [21] presents the development of rubber waste recycling, methods, characteristics, and improvements that contributed to the improvement of the above-mentioned properties of waste rubber. Emphasis is placed on the development of composites filled with material from used rubber tires. The use of rubber waste as fillers improved the mechanical and physical properties of the composites. In this way, composites with waste rubber fillers have created a new sustainable material and have the potential to be used in various fields [20,22].

Most research on recycled rubber composites shows that recycled rubber reinforced with microscale particles leads to the development of physical and mechanical properties of structures and also provides cheap and lightweight composites for several application areas. In addition, recycled rubber composites may be suitable for applications where high strength and high impact resistance are desired.

The latest research shows a positive effect of the addition of rubber or its derivatives on the impact and ballistic properties of layered composites. In the work in [23], the influence of the matrix type on the energy absorption by the composite was examined. Two types of matrix were used—rubber and thermosetting (epoxy). The results showed that the rubber matrix enhances the energy absorption of the fabric while maintaining the flexibility of the composite.

Composites based on epoxy resin have favorable strength and physicomechanical properties with low weight and simple manufacturing technology. The properties of these composites and the great possibilities of modifying their properties through the use of additives of various origins make them applicable in many industries [24,25,26,27].

The research analyses conducted earlier by the authors of this study show that fiber composites used for engineering structures in the form of multi-layer laminates are highly sensitive to impact loads [28,29]. In previous studies, the authors analyzed the effect of the type of resin used [28] and the effect of the percentage amount of rubber recyclate added and the way it was distributed on the mechanical properties of epoxy–glass composites [30,31]. The subsequent tests showed higher strength parameters of composites based on EPIDIAN^@^6 epoxy resin and with the use of a 5% addition of rubber recyclate in relation to the previously tested materials based on EPO 652 resin. The addition of rubber recyclate in the form of one, two, and three sandwich layers resulted in better strength parameters than composites with the addition of rubber modifier in the amount of 3, 5, and 7% randomly distributed in the matrix. The basic measurement results are presented later in this paper.

In most of the solutions used, rubber recyclate is used as a filler that acts as a flexible material. On this subject, work was carried out to determine how the percentage content and application in the layered composite affect the strength parameters of the newly produced materials. For the presented research, epoxy–glass composites with a limited addition of recyclate (maximum 5% of the weight of the composite) were produced and tested. The aim of the conducted research and analysis was to isolate a composite with the most favorable mechanical properties in terms of tensile strength, impact strength, and damage kinetics from several variants of test materials. The selected type of composite is to be used for further comparative research using a new production technology and the type of reinforcement and resin used is to be made with vacuum infusion technology.

## 2. Materials and Methods

Epoxy–glass composites were made using the following materials:(1)EM 1002/300/125 glass mat with irregular fiber distribution, weight 350 g/m^2^ [16];(2)EPIDIAN@6 epoxy resin^®^6 [32];(3)Hardener Z-1, an amine hardener (aliphatic amine) with the following properties [33]:

Viscosity at 25 °C: 20–30 (mPa s);

Density at 20 °C: 0.978–0.983 (g/cm^3^);

Amine number: min. 1100 (mgKOH/g).

(4)Rubber recyclate from the recycling process of car tires with granulation of 0.5 mm to 3 mm [34].

Table 3 shows the characteristics of the Epidian^®^ 6 epoxy resin used to make the composites.

Hardener Z-1 was added to all composites in the amount of 13 g/100 g of epoxy resin. Figure 1 shows the LAB-11-200 shaker from EKOLAB, used to obtain a fraction of rubber recyclate with a size of 0.5 mm to 1.5 mm, and the materials used to produce composite materials.

The test material was made as a sandwich composite containing a certain number of layers of glass mat, layers of rubber recyclate, and EPIDIAN^@^6 epoxy resin. All variants of the produced materials contained the same number of layers of glass mat (12) and the same percentage of modifier content in the form of rubber recyclate (5%) and EPIDIAN^@^6 epoxy resin (60%). The composition of the materials is described in Table 4, and the mass content of the components of the manufactured variants of the new materials is presented in Table 5.

To produce the materials, the manual lamination method was used using a brush, rollers, and molds with dimensions of 300 × 900 mm. In addition, the composite was pressed from above with a steel sheet with dimensions of 295 × 895 × 6 mm and weights (the method of load arrangement is shown in Figure 2 and Figure 3. The pressure value (675 N per 1 mold) was experimentally determined in such a way that there was no excessive seepage and leakage of the resin outside the mold.

In each of the variants of the composite, exactly the same amount of saturating agent, pressure value, time (7 days), and curing temperature (22 °C) were used. The only basic difference was the method of distribution of rubber recyclate layers in the composite (Figure 4).

### Planning and Conditions of the Experiment

Standardized test specimens for static tensile and impact tests were cut out of the panels made by water cutting. The samples were prepared in accordance with PKN standards: PN-EN ISO 179-1:2010E and PN-EN ISO 527-4_2000P. The cut samples were subjected to a static tensile test on a Zwick&Roell testing machine. Impact tests were carried out on a Charpy-type pendulum hammer RKP450 with TestXpert II software. Thanks to additional software, it was possible to determine not only impact strength, but also changes in bending force and registration of deflection over time. Figure 5 and Figure 6 show the dimensions and geometry of the samples used in the static tensile and impact tests.

Figure 7 shows the samples prepared for the static tensile test and Figure 8 shows the samples during static tensile tests. Figure 9 shows composite samples before the impact test and laboratory stand for the impact tests.

During the impact tests, impact values were recorded for a set of samples for four types of test materials. In addition, thanks to the owned software, the results of the analysis of changes in the value of the bending force, as well as the deflection of the samples in small time intervals, were recorded. Based on the obtained results, force–displacement diagrams were obtained, which were used to illustrate the energy expenditure necessary to destroy the sample in the area of elastic deformations.

The obtained results were used to perform a further comparative analysis of the possessed materials in terms of the above-mentioned parameters. In addition, the results allowed for the selection of the most advantageous variants of the tested composite materials.

## 3. Results and Discussion

### 3.1. Test Results Obtained in a Static Tensile Test

The basis for the further part of the research was the previously obtained results and analyses of measurements on samples made on the matrix of EPIDIAN^@^6 resin with random addition of rubber recyclate to the composite matrix in the amount of 3, 5, and 7% (marked, respectively, in this study as L3, L5, and L7) in relation to the comparative sample K0 without the addition of rubber recyclate. The obtained test results and analyses were included in the study [30]. For the purposes of this study, Figure 10, Figure 11 and Figure 12 show graphs from a static tensile test for five samples from three material variants.

Figure 13, Figure 14, Figure 15 and Figure 16 show graphs from the static tensile test of the tested composite samples Ko, K1, K2, and K3 obtained from the TestXpert II software of the Zwic&Roell testing machine. The list of parameters of the tested composite materials from the TestXpert II software is presented in Table 6.

The analysis of the obtained results allowed us to determine that the most favorable parameters in terms of strength were achieved for the *K1* material (Young’s modulus *E* decreased by 26.5%, *ε* increased by 17.9%, *σ_m_* decreased by 19.1%). The lowest parameter values were obtained by the *K3* material (Young’s modulus *E* decreased by 29.2%, *ε* increased by 9.2%, *σ_m_* decreased by 27.2%). The above parameter values were obtained in relation to the parameters of the comparative composite *K0* (without the addition of recyclate). The above-mentioned parameter values clearly indicate that the addition of recyclate for all variants caused a decrease in strength in the range of *σ_m_* and *E* and an increase in the deformation value *ε*. A summary of the relationships described above is presented in Table 7.

### 3.2. Test Results Obtained in the Impact Test

Impact strength measurements were made for four variants of materials and 12 samples of each type (K0, K1, K2, K3). Figure 17, Figure 18, Figure 19, Figure 20 and Figure 21 show the results of the impact test. Table 8 presents the values of the parameters regarding the impact strength of the tested materials.

Figure 22a–d show a graph of deflection versus force for the fabricated materials subjected to a dynamic Charpi test. The graphs in question illustrate the recording of the work of the elastic state (Ue) and the development of failure (Up) for one selected sample from each material variant.

Table 9 shows the average values of the maximum force *F_MAX_*, deflections *f*, work *W*, and impact strength *U* for all variants of the tested samples of four types of material.

The results obtained from the measurements of the impact strength of the sandwich composites showed the influence of the arrangement of the recyclate as sandwich layers (one, two, or three layers, 5%) on the impact strength of the new material and on the course of the destruction process of the composite samples during dynamic loads. The best impact properties were obtained for the *K2* composite. For a sample of this material, the maximum *F_MAX_* force of 3240 N caused a deflection of 1.85 mm. For the *K0* sample, without the addition of rubber recyclate, the maximum force of 2983 N caused a deflection of 1.52 mm. The impact strength values for the *K2* samples could be affected by the symmetry of the arrangement of the rubber recyclate layers relative to the longitudinal axis of the sample.

#### Statistical Analysis of the Obtained Results of Impact Strength Measurements

The statistical analyses carried out made it possible to define and isolate the most advantageous variants of the produced materials with the best strength parameters, taking into account the range of their anisotropicity. The use of statistical analysis is a very helpful tool in modern computational methods in the analysis of repeatability and reliability of the obtained measurement results, especially in the case of testing anisotropic materials.

The results of 12 measurements for four material variants (*K0*, *K1*, *K2*, *K3*) obtained in the impact test in the range of parameters (Average, Average ± std error, Average ± 1.96 * Std error) were subjected to statistical analysis.

Testing the normality of distributions with Shapiro–Wilk test

The Shapiro–Wilk test was used to analyze the results, as it is considered the best test to check the normality of the random variable distribution. The main advantage of this test is its high power, i.e., for a fixed level of statistical significance α, the probability of rejecting the hypothesis H_0_ if it is false is greater than in other tests of this type. For further statistical analysis, the following assumptions were made: α = 95% and the null and alternative hypotheses of the following form:

H_0_: The distribution of the examined feature is a normal distribution.

H_1_: The distribution of the examined feature is not a normal distribution.

Table 10 presents the obtained analysis values using the Shapiro–Wilk test for the tested composite materials.

All obtained results of p-probability values turned out to be greater than 0.05; therefore, there was no reason to reject the null hypothesis. The impact strength variables for all tested samples were normally distributed. Therefore, parametric tests were used to test differences.

Testing differences between pairs

The impact strength variables for all tested samples were normally distributed; therefore, the Student’s *t*-test for related samples was used to test the differences. Material K0 was taken as a standard sample. The following hypotheses were assumed:

The null hypothesis of no differences between population means;

H_0_: μ1 = μ2.

The alternative hypothesis about the occurrence of differences between population means was as follows:

H_1_: μ1 ≠ μ2.

Table 11 presents the obtained values of the statistical analysis of the impact strength measurements of the tested materials using the Student’s *t*-test.

As a result of the analysis of the data obtained, it was found that for all the results obtained, the probabilities *p* were less than 0.05; therefore, there was no basis for accepting the null hypothesis. Significant statistical differences occurred between pairs of impact strength variables for all tested samples. Figure 23a–c show the box–whisker plots for the tested materials *K0*, *K1*, *K2*, and *K3* in the analysis range mean, mean ± standard error, and mean ± 1.96 standard error.

ANOVA difference testing

As shown in the previous section of testing, there were significant statistical differences between all samples in the configurations of reference sample *K0* and samples *K1*, *K2*, and *K3*, and there was a significant change in impact strength. However, it was decided to check whether there are significant statistical differences between the tested samples corresponding to the tested types of materials treated as independent samples. The impact strength variables for all tested samples were normally distributed; therefore, the ANOVA test was used to test the differences. The independent-sample ANOVA test is an extension of the Student’s *t*-test for more than two samples. The following hypotheses were defined:

Null hypothesis of no differences between population means H_0_: μ1 = μ2 = μ3 = μ4; An alternative hypothesis about the occurrence of differences between population means H_1_: μ1 ≠ μ2 ≠ μ3 ≠ μ4.

The Brown–Forsythe test was used to check the equality of variances, as it is less sensitive than the Levene test to the failure to meet the assumption of normal distribution. The received value *p* = 0.674862427 was greater than the composite minimum significance threshold *p_v_* = 0.05, so the assumption of equality of variances was met. Therefore, ANOVA tests were performed to assess the significance of the average variation in impact strength.

The value of the calculated test probability *p* = 0.00 allowed for the rejection of the null hypothesis assuming no differentiation of average values. As a result, not all sample groups came from one population, and this was the basis for making further “posterior” (post hoc) comparisons.

In total, 12 comparisons were made for all material configurations. The results of the NIR test and the chi-square median test (whiskers frame) are presented in Table 12; they showed that there were nine comparisons corresponding to the types of material between which there were significant statistical differences, and there was one comparison for which there were no significant statistical differences between the tested materials, i.e., K1&K3’s calculated value *p* = 0.071445 was greater than the assumed minimum significance threshold *p_v_* = 0.05).

Figure 24a,b show the box–whisker plots for the tested materials *K0*, *K1*, *K2*, and *K3* in terms of mean, standard error, and standard deviation.

Based on the statistical analysis of the impact strength results, it can be concluded that the distributions of the impact strength values for all samples are normal distributions, which allows the use of statistical tests to compare the results of experimental tests.

The obtained *p*-values of pairwise comparisons and multiple ANOVA comparisons showed that there were significant statistical differences for all samples (calculated *p* = 0.004020 for K0&K1, *p* = 0.000969 for K0&K2, and *p* = 0.000032 for K0&K3, all smaller than the assumed *p_v_*= 0.05). In connection with the above, we can conclude that the addition of rubber recyclate as a spacer layer significantly changed the strength parameters of samples *K1*, *K2*, and *K3* in relation to sample *K0.*

The conducted statistical analyses confirmed the best results of measuring the impact strength values obtained from the RKP450 TestXpert II Charpy Hammer software for the K2 sample. At the same time, the cluster analysis in the range (mean, mean ± standard error, mean ± standard deviation) showed that the most similar results were obtained for the K3 sample. Measurement results for this material are the most repeatable, which indicates its greater homogeneity.

### 3.3. Measurements Using the SEM Electron Microscope

The Zeiss EVO MA 15 scanning electron microscope was used to analyze the structures of the composites described in the article (Figure 25). The purpose of the structure analysis was to initially determine the impact of the rubber modifier on the arrangement and adhesion of the components of the variants of the *K1*, *K2*, and *K3* research materials produced.

In earlier studies, for the *L3*, *L5*, and *L7* material variants, the interior of the tested samples was X-rayed using the ZEISS METROTOM 6 Scout tomograph and the course of deformations in the samples was analyzed using the GOM Suite 2021 software [30]. The same analysis is planned for the materials presented in this study.

The performance of the tests in question will be helpful in the analysis of the porosity of these materials. Specimens for observation under the microscope were produced using plate fragments from which samples used in the static tensile test and impact test were cut. In the process of the preparation of cross-sections, abrasive papers were used (320, 800, and 1200 grit), and cross-sections were then polished with a polishing slurry with a grain size of 3 μm.

Figure 26a–d show pictures of microstructures of cross-sections of samples from the analyzed composite materials.

The analysis of the obtained images of the SEM structures showed a clear influence of the recyclate addition on the differences in the structure of all the tested materials in relation to the comparative sample *K0*. Observation of cross-sections showed that in the case of the *K3* variant, the adhesion of the recyclate to the resins and glass fibers was greater than the analyzed variants *K1* and *K2*, with visible air pores. At the same time, the influence of the rubber modifier on the internal structures of variants *K1*, *K2*, and *K3* is noticeable. The analysis of the structures shown in Figure 26b–d allows us to conclude that air pores were formed in the vicinity of the rubber modifier and on the glass mat–resin border, visible as dark voids in the photos. Air pores will have a significant impact on the strength parameters, reducing their value.

## 4. Conclusions

The tests carried out, with particular emphasis on the strength parameters obtained in the static tensile test and the impact test, were aimed at determining the optimal parameters of the tested materials and separating the most advantageous variants depending on the planned application. The tests showed that the addition of rubber recyclate and the way it was distributed in the layers of the composite had a significant impact on its strength properties. Based on the analysis of the results, it can be assumed that the most favorable strength parameters in terms of the static tensile test were shown by the *K1* material samples, but they were lower in relation to the *K0* base material. Rubber recyclate decreased the values of most strength parameters; only deformation *ε* for samples with recyclate had higher values.

The analysis of the results from the impact test showed that the most favorable parameters in terms of impact strength were shown by the material that obtained impact resistance *U* at a higher level than the *K0* base material. The statistical analysis performed for the obtained impact strength measurements confirmed the experimental results.

Due to the dynamics of the impact strength test and based on the parameters readable on the measuring stand and the Zwick Roell software, it was not possible to obtain accurate information on the impact of the recyclate addition on the cracking mechanism. Due to the values of the obtained results and the elastic properties of the modifier in the form of rubber recyclate, it can be concluded that the addition of recyclate in selected distribution variants and the percentage of the additive delay the sample breaking process. The greatest delay may occur for the *K2* variant of the sample with two layers of recyclate.

Due to the properties of layered composites, it is possible to model and design materials by adapting them to specific applications and manufacturer requirements. The conducted research shows that the use of recycled materials enables the creation of new, environmentally friendly materials with more favorable performance and economic parameters.

At a later stage of the study of the material variants in question, the authors plan to study the effect of temperature on the mechanical reaction and the phenomena occurring in the material at different temperatures. A very important aspect will also be the analysis of thermal conductivity, determination of the porosity of materials, dynamic mechanical analysis, and electrical conductivity.

The intention of further research will also be to obtain a variant of the material with increased vibration damping and acoustic insulation properties in relation to the same composite without the use of rubber recyclate. These studies will be focused on the assessment of vibro-isolation and reduction in noise transmission in terms of the use of these materials for structures, e.g., low-noise vessels or yachts.

In addition, at a later stage of research, the authors will abandon the manual lamination process and create new, perspective variants of composites in the manufacturing process using the vacuum infusion method and a vacuum bag.

## Figures and Tables

**Figure 1 polymers-15-03374-f001:**
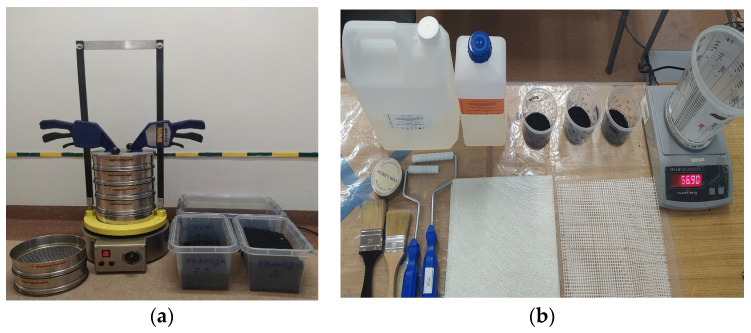
(**a**) EKOLAB sieve shaker LAB-11-200. (**b**) Materials used in the composite manufacturing process.

**Figure 2 polymers-15-03374-f002:**
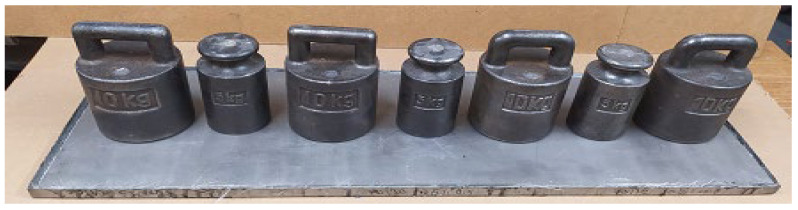
Production of research materials K3.

**Figure 3 polymers-15-03374-f003:**
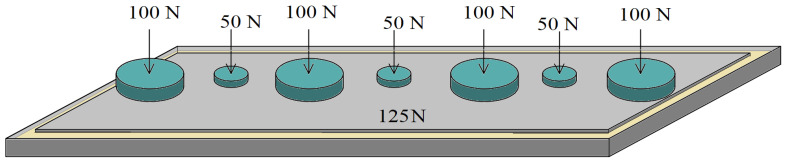
Load diagram during the production of composite materials.

**Figure 4 polymers-15-03374-f004:**
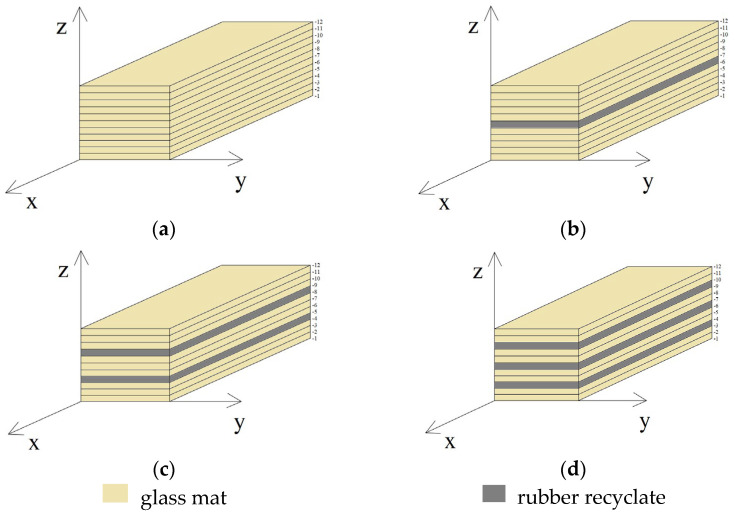
(**a**) Scheme of laying layers of glass mat and rubber recyclate in the following materials: (**a**) K0; (**b**) K1; (**c**) K2; (**d**) K3.

**Figure 5 polymers-15-03374-f005:**
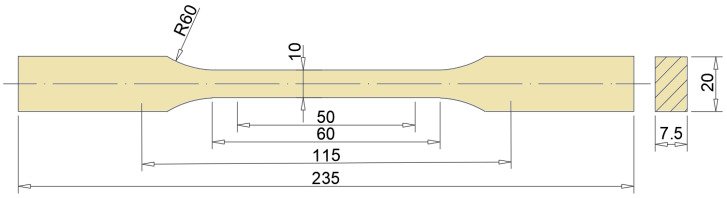
Shape and dimensions of samples intended for testing a tangential tensile test.

**Figure 6 polymers-15-03374-f006:**
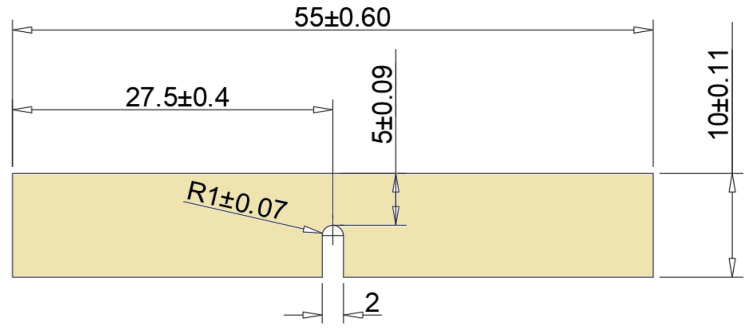
Shape and dimensions of samples for impact strength testing.

**Figure 7 polymers-15-03374-f007:**
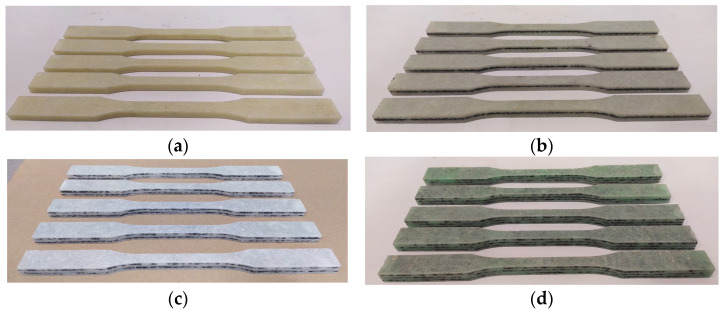
Specimens for static tensile test of composite (**a**) K0; (**b**) K1; (**c**) K2; (**d**) K3.

**Figure 8 polymers-15-03374-f008:**
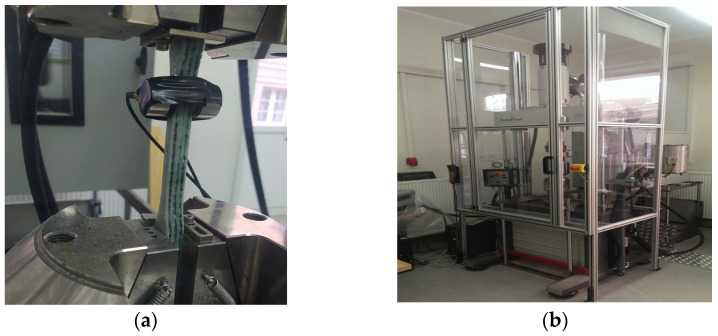
(**a**) K3 composite samples during the static tensile test. (**b**) Stand for testing a static tensile test.

**Figure 9 polymers-15-03374-f009:**
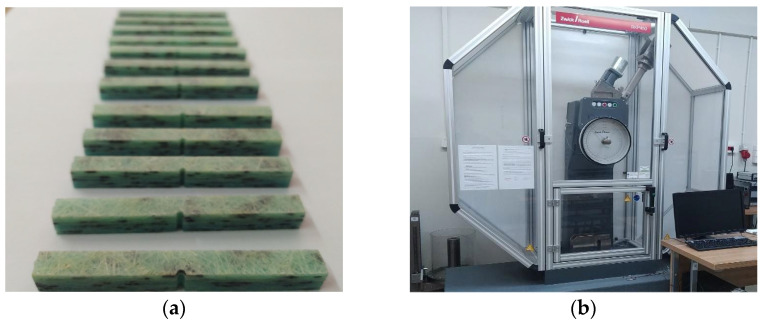
(**a**) K3 composite specimens for impact tests. (**b**) RKP450 Charpy pendulum hammer.

**Figure 10 polymers-15-03374-f010:**
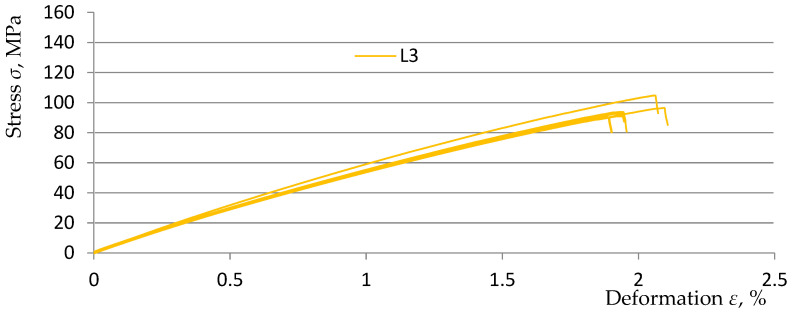
Stress–strain diagram for samples of L3 material [30].

**Figure 11 polymers-15-03374-f011:**
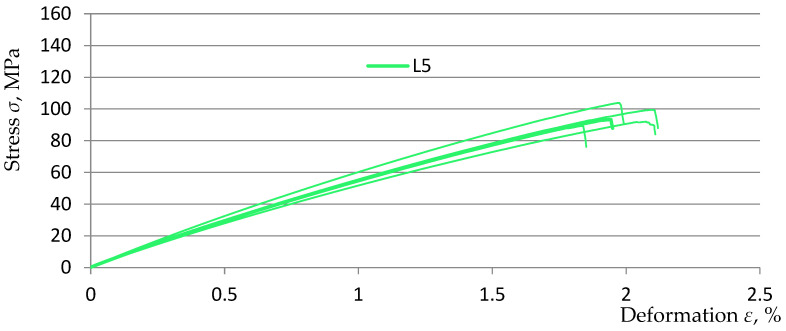
Stress–strain diagram for samples of L5 material [30].

**Figure 12 polymers-15-03374-f012:**
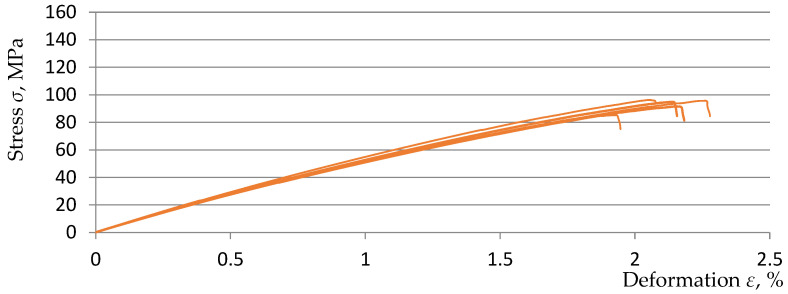
Stress–strain diagram for samples of L7 material [30].

**Figure 13 polymers-15-03374-f013:**
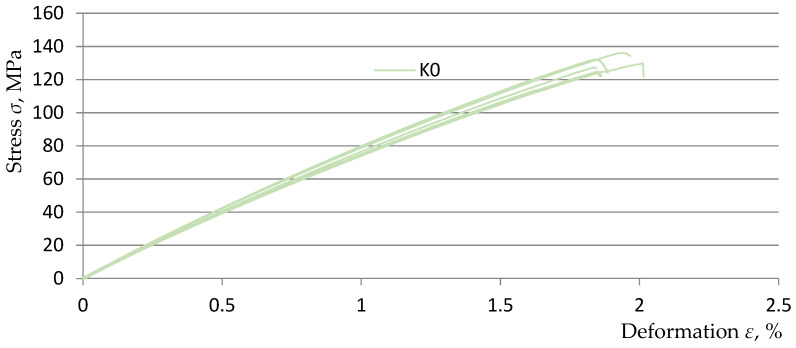
Static tensile test diagrams for 5 K0 composite samples.

**Figure 14 polymers-15-03374-f014:**
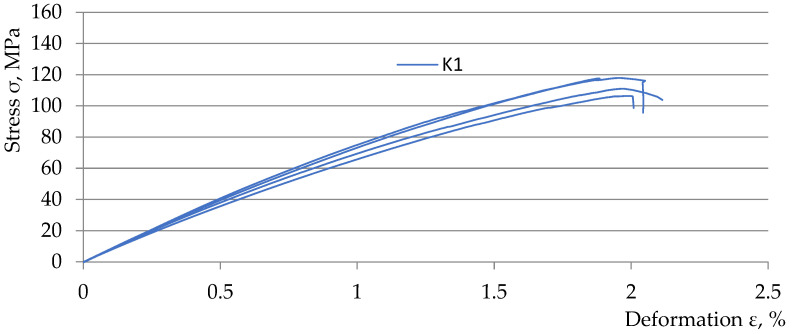
Static tensile test diagrams for 5 K1 composite samples.

**Figure 15 polymers-15-03374-f015:**
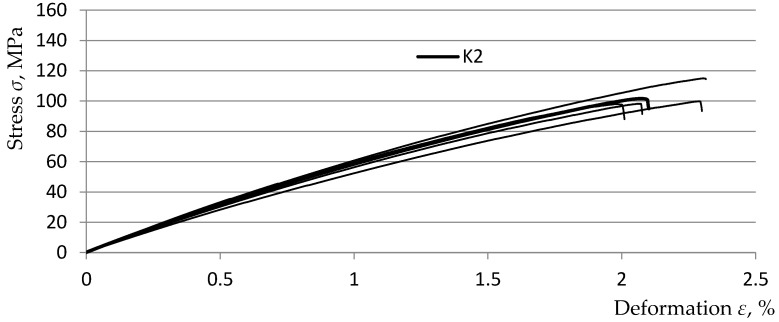
Static tensile test diagrams for 5 K2 composite samples.

**Figure 16 polymers-15-03374-f016:**
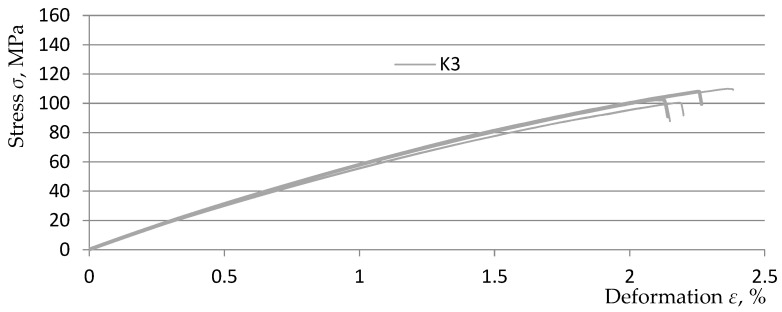
Static tensile test diagrams for 5 K3 composite samples.

**Figure 17 polymers-15-03374-f017:**
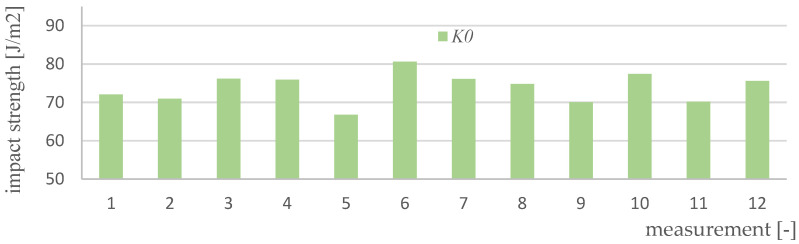
Results of impact strength measurements for K0 material samples.

**Figure 18 polymers-15-03374-f018:**
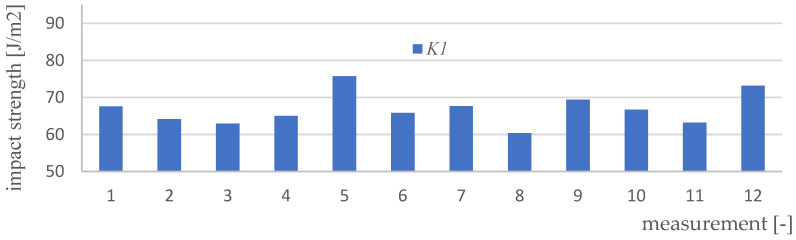
Impact test results for K1 material samples.

**Figure 19 polymers-15-03374-f019:**
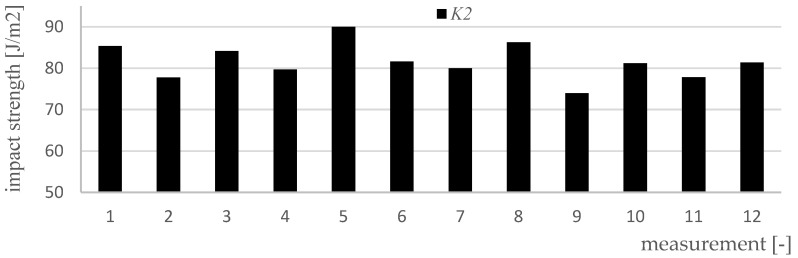
Results of impact strength measurements for K2 material samples.

**Figure 20 polymers-15-03374-f020:**
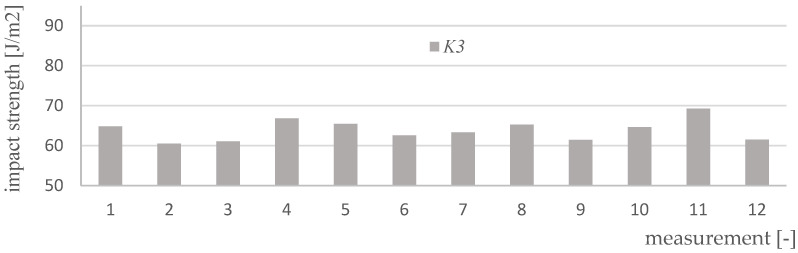
Impact test results for K3 material samples.

**Figure 21 polymers-15-03374-f021:**
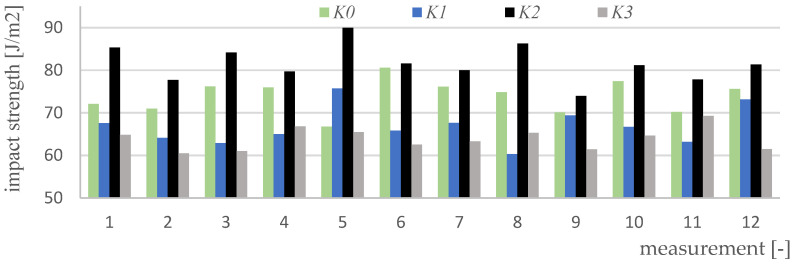
A summary of the results of impact strength measurements for all variants of the tested materials.

**Figure 22 polymers-15-03374-f022:**
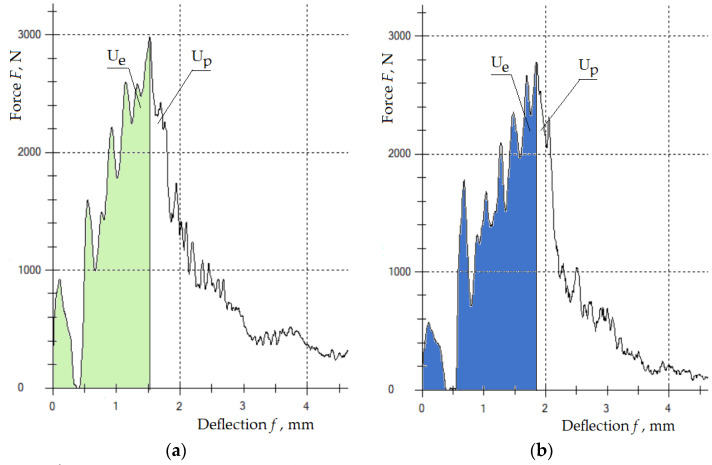
Force–deflection diagram *F(f)* (**a**) for sample *K0*; (**b**) for sample *K1*; (**c**) for sample *K2*; (**d**) for sample *K3*.

**Figure 23 polymers-15-03374-f023:**
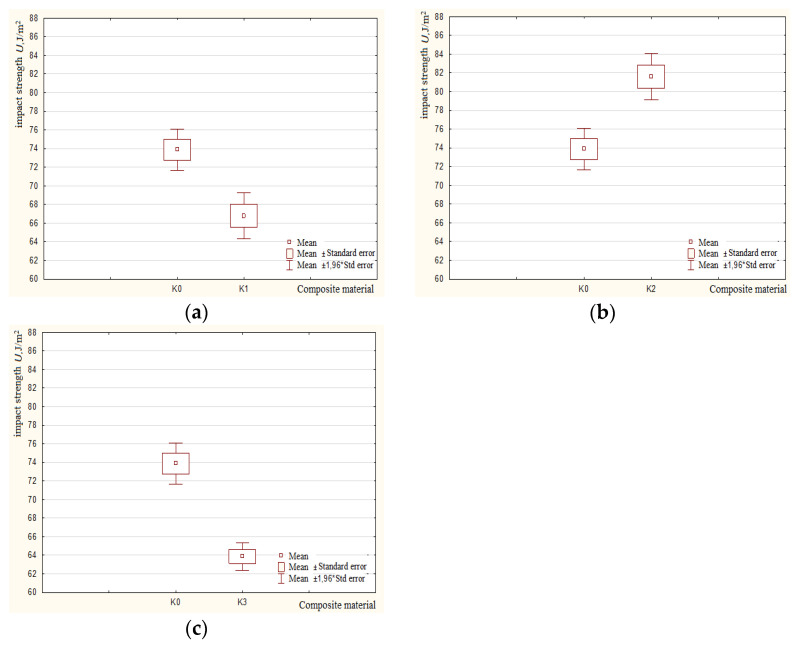
Box–whisker plots for the tested pairs of variables (Average, Mean ± Standard Error, Mean ± 1.96*Standard Error) and (Average, Mean ± Standard Error, Mean ± Standard) (**a**) for sample *K0* vs. sample *K1*; (**b**) for sample *K0* vs. sample *K2*; (**c**) for sample *K0* vs. sample *K3*.

**Figure 24 polymers-15-03374-f024:**
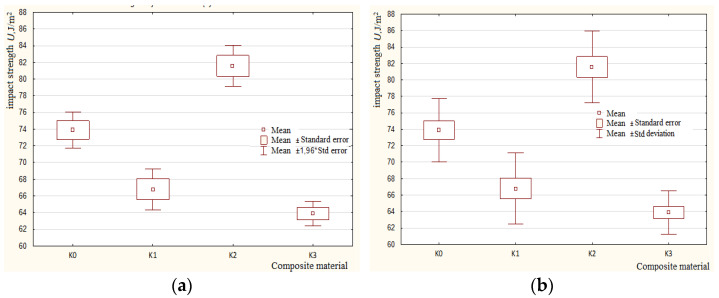
Box–whisker plots for tested materials. (**a**) Mean, Mean ± std error, Mean ± 1.96*std error; (**b**) Mean, Mean ± Standard Error, Mean ± Standard.

**Figure 25 polymers-15-03374-f025:**
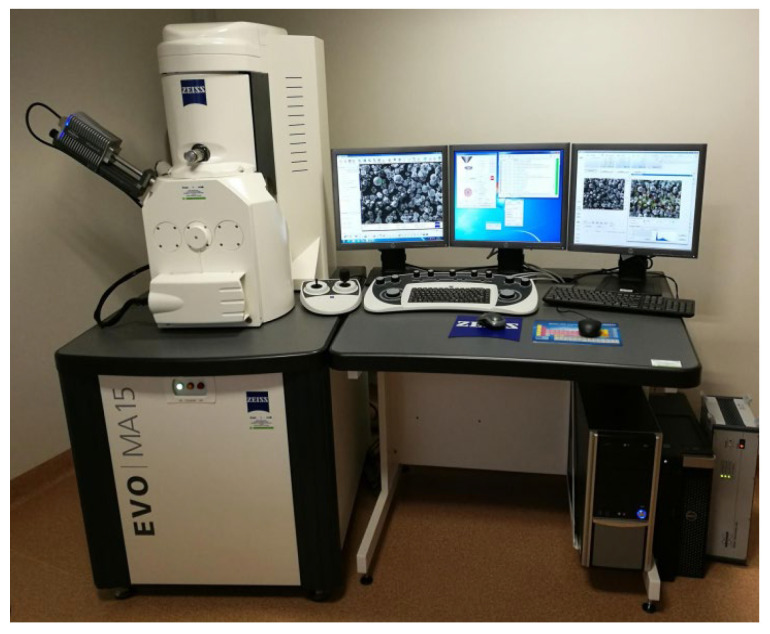
Scanning electron microscope Zeiss EVO MA 15 was used to observe the microstructure of the tested materials.

**Figure 26 polymers-15-03374-f026:**
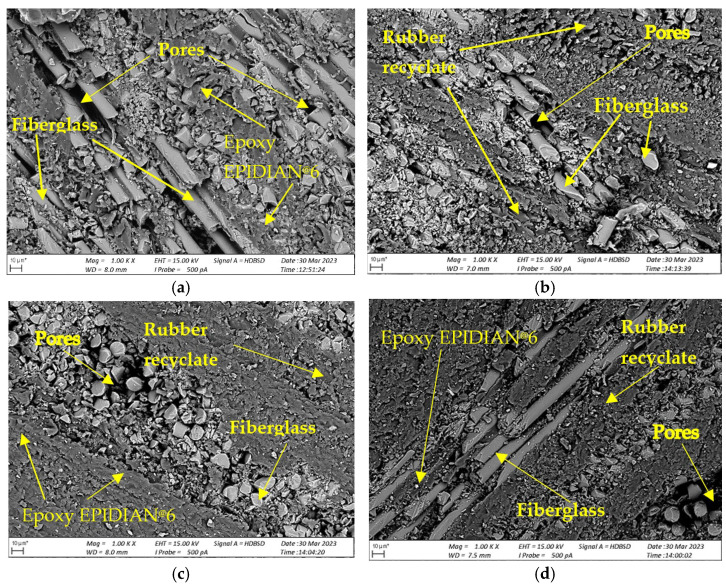
SEM structures (1000×) for (**a**) comparative sample *K0*; (**b**) *K1* composite; (**c**) *K2* composite; (**d**) *K3* composite.

**Table 1 polymers-15-03374-t001:** Supply of used tires in EU countries and Poland, taking into account the development of the automotive industry [10].

Country	The Number of Residents, Million	Number of Cars/ 1000 Inhabitants	Tire Wear, Thousands of Tons/Year
France	58	419	354
Germany	81	399	603
Italy	58	520	330
Anglia	58	379	378
Spain	39	321	202
Sweden	8.7	213	119
Poland	38.7	213	119

**Table 2 polymers-15-03374-t002:** Quantity of tires produced and introduced to the Polish industry [1].

Year	Tire Production, Thousands	Number of Tires Entered for the Polish Market, Thousands of Tons	Recycled Tires, Thousands of Tons
2015	46.715	222.2	175.3
2016	47.284	244.7	192.0
2017	46.271	281.1	211.8

**Table 3 polymers-15-03374-t003:** EPIDIAN^®^6 epoxy resin characteristics [32].

Parameter	Unit	Value
Epoxy number	[Mol/100 g]	0.510–0.540
Density at 25 °C	[g/cm^3^]	1.17
Viscosity at 25 °C	[mPa s]	1000–1500
Gel time 100 g in 20 °C	[min]	20
Curing time in 20 °C	[days]	7

**Table 4 polymers-15-03374-t004:** Description of the composition of the tested variants of materials.

Composite	Composition Description
*K0*	12 layers of glass mat soaked with epoxy resin, without the addition of rubber recyclate.
*K1*	12 layers of glass mat soaked with epoxy resin. Rubber recyclate is added to the composite in the amount of 5% of the total weight of the composite as 1 interlayer layer, between the 6th and 7th layer of the glass mat.
*K2*	12 layers of glass mat soaked with epoxy resin. Rubber recyclate is added to the composite in the amount of 5% of the total weight of the composite as 2 interlayer layers, between the 4th and 5th and the 8th and 9th layers of the glass mat.
*K3*	12 layers of glass mat soaked with epoxy resin. Rubber recyclate is added to the composite in the amount of 5% of the total weight of the composite as 3 interlayers, between the 3rd and 4th, 6th and 7th, and 9th and 10th layers of the glass mat.

**Table 5 polymers-15-03374-t005:** Mass content of the components of the epoxy–glass composite with the addition of rubber recyclate as a spacer layer, made by hand lamination.

Composite	Mat Layers	Resin Content, %	Glass Mat Content, %	Number of Layers of Rubber Recyclate	Recyclate Content, %
*K0*	12	60%	40%	0	0%
*K1*	12	60%	35%	1	5%
*K2*	12	60%	35%	2	5%
*K3*	12	60%	35%	3	5%

**Table 6 polymers-15-03374-t006:** Strength parameters obtained from the TestXpert software of the Zwick Roell machine for composite materials.

Material	*σ_m_* [MPa]	*ε* [%]	*E* [MPa]
K0	136	1.95	8742
K1	111	1.97	7965
K2	110	2.30	6426
K3	99	2.13	6185

**Table 7 polymers-15-03374-t007:** Values of percentage changes in the strength parameters of composites with the addition of rubber recyclate in relation to composite K0 (data from Test Xpert II software).

Material	*σ_m_* [MPa]	*ε* [%]	*E* [MPa]
*K1*	−18.4	1.0	−8.9
*K2*	−19.1	17.9	−26.5
*K3*	−27.2	9.2	−29.2

**Table 8 polymers-15-03374-t008:** Values of obtained impact strength measurements for composite materials.

Material	*K0*	*K1*	*K2*	*K3*
Impact strength, *U*, J/m^2^	72	68	85	65
	71	64	78	60
	76	63	84	61
	76	65	80	67
	67	76	90	65
	81	66	82	63
	76	68	80	63
	75	60	86	65
	70	69	74	61
	77	67	81	65
	70	63	78	69
	76	73	81	62
Mean	74	67	82	64

**Table 9 polymers-15-03374-t009:** Average results of the impact tests performed on the tested variants of composites.

Composite/Parameter	*F_MAX_* [N]	*f* [mm]	*W* [J]	*U* [J/m^2^]
*K0*	2983	1.52	5.06	74
*K1*	2774	1.85	4.82	67
*K2*	3240	1.81	4.88	82
*K3*	2889	1.67	5.56	64

**Table 10 polymers-15-03374-t010:** Calculated values of the Shapiro–Wilk test statistic for the tested samples.

Sample Name	Value *p*
*K0*	0.730370
*K1*	0.649216
*K2*	0.986275
*K3*	0.575654

**Table 11 polymers-15-03374-t011:** Computed test statistic values of the Student’s *t*-test for paired samples for the tested samples.

Pair of Variables	*p*-Value
K0&K1	0.004020
K0&K2	0.000969
K0&K3	0.000032

**Table 12 polymers-15-03374-t012:** Calculated independent-sample post hoc test statistic values for test samples.

	Test NIR. P-Probabilities for Post Hoc Tests (2-Sided).			
Material	*K0*	*K1*	*K2*	*K3*
*K0*	-	0.000050	0.000014	0.000000
*K1*	0.000050	-	0.000000	0.071445
*K2*	0.000014	0.000000	-	0.000000
*K3*	0.000000	0.071445	0.000000	-

## Data Availability

All the data used in the manuscript has been cited properly.

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
