# Peer review of "Analysis of the Impact of Rubber Recyclate Addition to the Matrix on the Strength Properties of Epoxy–Glass Composites"

_polymers, 2023, doi:10.3390/polym15163374_

Round 1

Reviewer 1 Report

The paper entitled " Analysis of the Influence of Rubber Recyclate Addition to the Warp on the Strength Properties of Epoxy-Glass Composites’’ deals with the development of recycling rubber waste into composite materials. The scope of the work is very interesting and will provide significant scientific insight in the field. However, according’s reviewer’s opinion there are several flows and some parts need significant improvement in order to be publishable. The major points are presents below:

1.      The introduction needs to be enriched along with the corresponding literature. The scope and the contribution of the work should be underlined and presented in a more straightforward way.

2.      Would be possible to prepare more samples in different concentrations?

3.      Error bars in figure 8 and  9 and more than necessary. In order to be possible to evaluate their repetition. More samples in each concentration should be prepared in order to be able to study their behavior.

4.      Could temperature be also a parameter for their mechanical response? A similar study in different temperature level would provide significant insight regarding their behaviour.

5.      Thermal conductivity? Some thermal analysis results is also necessary

6.      Electrical conductivity or even dynamic mechanic analysis could also provide important information.

Moderate editing of English language required

Author Response

The article entitled "Analysis of the influence of the addition of rubber recyclate to the matrix on the strength properties of epoxy-glass composites" concerns the development of recycling of rubber waste into composite materials. The scope of work is very interesting and will provide relevant scientific insight into the field. However, according to the reviewer, there are several flows, and some parts need significant improvement before they can be published. The main points are set out below:

  1. The introduction should be enriched with appropriate literature. The scope and contribution of the work should be emphasised and presented in a more direct way.

Response 1: The authors have enriched the content of the Introduction with new, new literature items more related to the scope and subject matter of research and analysis.

The introduction has been improved in terms of presenting the scope and contribution of work in a clearer and more direct way.

  1. Would it be possible to prepare more samples in different concentrations?

Response 2 : The researchers additionally included in the study information on previously conducted analyzes of composite materials made in the same technology and from the same materials, but differing in the method of arrangement and different content of recyclate in the composite.

  1. Error bars on Fig. 8 and 9 and more than necessary. To be able to assess their repeatability. More samples should be prepared at each concentration to study their behaviour.

Response  3 : The authors increased the number of tested samples in the Figures in order to enable the assessment of the repeatability of measurements. The authors in previous studies referred to materials made and tested with different percentages of rubber recyclate in the composite. Basic data on this subject are presented in Figures 10 , 11 and 12 and publication [30].

  1. Can temperature also be a parameter of their mechanical reaction? A similar study at different temperatures would provide important insights into their behavior.
  2. Thermal conductivity? The results of the thermal analysis are also necessary
  3. Electrical conductivity and even dynamic mechanical analysis can also provide important information.

Response for points 4,5,6 : Due to the wide range and large number of variants of the manufactured composite materials, at this stage the investigators limited themselves to the following scope:

  • Static tensile test,
  • Toughness
  • Hardness measurement,
  • Analysis of the course of deformation during the static tensile test using the ARAMIS system and GOM Suite 2021,
  • ZEISS METROTOM 6 Scout tomograph scanning to obtain information about real cross-sections,
  • Acoustic tests,
  • Statistical analysis of the results obtained

Due to the large extensiveness that are the subject of previous, current and subsequent studies.

Thank you very much for paying attention to such important parameters for newly manufactured materials as the influence of temperature on mechanical reaction, thermal conductivity, electrical conductivity and dynamic mechanical analysis, which will certainly provide further key information on the properties of newly manufactured materials. Because researchers need to prepare for the further proposed scope of research, the time given to improve the post-review publication is too short to prepare, analyse and develop the studies and their results. The proposed new areas of research will certainly be taken into account and carried out in further research work.

Comments on English:  Quality Moderate English editing required

Response: In the field of correction of the English language, the authors will turn to the Editorial Board of the journal

Reviewer 2 Report

The article: “Analysis of the Influence of Rubber Recyclate Addition to the Warp on the Strength Properties of Epoxy-Glass Composites”, includes the development of sandwich layer rubber recyclate obtained from worn car tires with epoxy-glass composites and an evaluation of the strength properties.

After the review, I have the following comments for the authors to improve the article.

1.         The number of references needs to be increased.

2.      The basis of the selection of epoxy-glass composites as a precursor is not given in the materials. What were the base polymer composites considered by the authors and how epoxy-glass composites were selected?

3.    In the layup process, the porosity is inevitable and it deteriorates the properties of the composites. Why porosity was not measured? The impact of pores on the strength and the degree of anisotropy in epoxy composites is quite significant and it cannot be ignored. Therefore, it should be included in the results.

4.       In Fig. 12, pores are clearly identified in the SEM images, please annotate them and revise the figure.

5.       What is the effect of rubber recyclate on the fracture mechanism? Does it delay the fracture or accelerate it? Please include it in the discussion.

                    6.       Why authors considered the wrap? Why the development of a                                  hybrid composite with a mix of glass fiber and rubber recyclate was not                        explored? Give reasons. 

Moderate editing of English language required

Author Response

The article: "Analysis of the influence of the addition of rubber recyclate to the matrix on the strength properties of epoxy-glass composites" includes the development of a layered rubber recyclate obtained from used car tires made of epoxy-glass composites and the assessment of the strength of the property.

After the review, I have the following notes for the authors to improve the article.

  1. The number of references should be increased.

Response 1: The authors have enriched the content of the Introduction with numerous, new literature items more related to the scope and subject matter of the research and analysis.

  1. The materials do not provide a basis for the selection of epoxy-glass composites as precursors. What base polymer composites did the authors consider and how were epoxy-glass composites selected?

Response 2:  The authors have included information on the beneficial properties of epoxy-glass composites, with particular emphasis on the possibility of modifying mechanical properties by adding new modifying components. The researchers focused on the use of various recycled materials and the assessment of their impact on the properties of sandwich composites based on polyester and epoxy resins and glass mat with their addition. The current research is their continuation, i.e. the modifying factor has been changed, while the base has remained unchanged. The researchers additionally included in the study information on previously conducted analyzes of composite materials made in the same technology and from the same materials, but differing in the method of arrangement and different content of recyclate in the composite.  Due to the acquisition of a new laboratory station for research work, enabling the production of composites by vacuum infusion and vacuum lamination, the authors began to perform and study new types of composites made using other material and technological variants, which will be presented in subsequent studies.

  1. In the lay-up process, porosity is inevitable and degrades the properties of composites. Why wasn't porosity measured? The influence of pores on the strength and degree of anisotropy of epoxy composites is quite significant and cannot be overlooked. Therefore, this should be taken into account in the results.

Response 3:  Due to the wide range and large number of variants of the manufactured composite materials, at this stage the investigators limited themselves to the following scope:

  • Static tensile test,
  • Toughness
  • Hardness measurement,
  • Analysis of the course of deformation during the static tensile test using the ARAMIS system and GOM Suite 2021,
  • ZEISS METROTOM 6 Scout tomograph scanning to obtain information about real cross-sections,
  • Acoustic tests,
  • Statistical analysis of the results obtained

Due to the large extensiveness that are the subject of previous, current and subsequent studies.

Thank you very much for paying attention to the influence of porosity on the properties of composites. Due to the need for preparation and short time to improve the article after the review, the proposed research will be done and presented as soon as possible. At the same time, the authors in the article added information on the tests carried out with the ZEISS METROTOM 6 Scout tomograph in order to obtain information about real cross-sections taking into account the anisotropic character and voids in the cross-sections. Due to the breadth of the current publication, the results in this area and porosity will also be included in subsequent studies.

  1. In Fig. 12, the pores are clearly marked on the SEM images, please annotate them and correct the drawing.

Response 4:  The authors corrected the drawing taking into account and describing examples of voids in the material (pores) in SEM images.

  1. What is the effect of rubber recyclate on the fracture mechanism? Delays the fracture or accelerates? Please include it in the discussion.

Response 5:  Due to the dynamics of impact testing and based on the parameters that can be read on your measuring station and Zwick Roell software, we are not able to obtain accurate information about the time, only the readable parameters are: the values of the maximum force FMAX, deflection f, work W and impact strength U. Due to the values of the obtained results and the elastic properties of the modifier in the form of rubber recyclate, we can conclude,  that the addition of recyclate in selected distribution variants and the percentage of the additive delays the process of breaking the sample. The greatest delay may occur for the K2 sample variant (with 2 layers of recyclate).

  1. Why did the authors consider resin? Why hasn't the possibility of developing a hybrid composite with a mixture of glass fibre and rubber recyclate been explored? Justify.

Response 6:  Due to the fact that the tested composites were produced by manual lamination using a brush and roller, it would be difficult to make a composite of one thickness and structure from mixed glass fibres and rubber recyclate. In addition, with the proposed method of pressure, the resin would be affected and heterogeneous distribution of fibers and recyclate in the composite would occur. It is very difficult to evenly distribute  the loose fibers  , as the fibers stick to the brush and the surface of the roller. Thank you very much for drawing attention to the possibility of making this type of composite, which will be possible using the vacuum infusion method currently used in our new material variants.

In the field of correction of the English language, the authors will turn to the Editorial Board of the journal

Reviewer 3 Report

Journal: Polymers (ISSN 2073-4360)

Manuscript ID polymers-2463992

Type: Article

Title

Analysis of the Influence of Rubber Recyclate Addition to the Warp on the Strength Properties of Epoxy-Glass Composites

Abstract:

Components present but may need minor writing for readability

• Subject matter is original and important

Introduction:

Minor rewriting; the hypothesis is clearly presented and is supported by the text

• References are adequate

Materials and methods:

Adequately written, although writing could be polished; minor typos/grammar/ punctuation errors

• Description of procedures needs minor clarification(clearly remediable)

Line 166: reference?

Figure 5: add a scale bare

Line 177: reference?

Figure 7 a, improve and add a scale bar

Results:

Statistical significance of findings not stated

Figure 8, is this the average of the five measurements?

Can figure 8 and 9 be combined?

Table 6: “Kompozyt”?

Figure 10, try to use the same font size for all graphs, Use the same scales

Statistical significance of findings not stated, is this an average measurement?

Discussion:

Statements, goals, and conclusions are not linked and are not clearly supported by data; major rewriting could address this, Why this complex analysis, explain. Be more concise.

Conclusion:

Some study implications and/or limitations are missing or not clearly presented

• Study has the potential to advance knowledge if paper is rewritten and key components clearly presented.

Adequately written, although writing could be polished; minor typos/grammar/ punctuation errors

Author Response

Abstract:

Components present, but may need to be finely written for readability

  • The subject matter is original and important

Admission:

Minor rewriting; The hypothesis is clearly presented and supported by the text

  • References are relevant

Materials and methods:

Properly written, although the writing can be refined; minor stylistic/grammar/punctuation errors

  • Description of procedures requires a small explanation (clearly repairable)

Response: The authors corrected the description of materials and methods as recommended. A table on the composition of materials has been added. Minor tweaks have been made to the text to make it more coherent and readable.

Line 166: reference?

Response: The authors have corrected the relevant reference.

Figure 5: Add an uncovered scale

Line 177: reference?

Response: The authors corrected references.

Response: The authors correct and add scale bar on the graphs.The authors added drawings concerning the geometrical dimensions of the tested samples in the static tensile test and impact test for all variants of the tested materials. Incorrect references have been corrected. The impact scale bars for all material variants have been standardized and improved.

Results:

The statistical significance of the results is not given

Figure 8, is this the average of five measurements?

Response: Figure 8 shows examples of single results out of 10 made for each variant. The authors modified the drawings by including 5 results for each variant of the material in order to allow the assessment of the repeatability of the obtained measurements.

Can Figures 8 and 9 be combined?

Response: The authors have placed other, new variants illustrating the studied materials and it will be difficult to combine them into a legible whole.

Table 6: "Composite"? 

Response: The authors have corrected the translation in Table 6.

Figure 10, try to use the same font size for all charts, use the same scales

The statistical significance of the results is not given, is it an average measurement?

Response: The authors corrected the graphs, standardized fonts and scales. The diagrams are selected, developed examples from the Zwick Roell impact test bench  program for each of the tested variants.

The results of the statistical analysis refer to the parameters presented in the Figures (in the range of Mean, Mean±Error std  , Mean±1.96*Error std). All statistical analyses concerned only the impact test, 12 measurement results for each of the material variants were used for the analyses. The above has been corrected and described in the article in accordance with the recommendation.

Discussion:

Statements, objectives and conclusions are not related to each other and are not clearly supported by data; The main rewriting could solve this problem, why this complex analysis, explain. Be more concise.

Response: The authors have organized and modified the organization of the article. They combined as a coherent whole the results of measurements, analyses and discussions.

Request:

Some of the implications and/or limitations of the study are missing or not clearly presented

  • The study has the potential to expand knowledge if the document is rewritten and the key elements are clearly presented.

 Response: The authors reorganized the content of the conclusions, indicating the key dependencies resulting from the obtained results and indicating further stages and directions of research development.

Comments on the quality of the English language

Properly written, although the writing can be refined; minor stylistic/grammar/punctuation errors

In the field of correction of the English language, the authors will turn to the Editorial Board of the journal

Round 2

Reviewer 1 Report

The authors have addressed successfully the reviewer's comments, thus it is acceptable for publication in this present form.

Moderate editing is required.

Reviewer 3 Report

Nice work, no remarks